# BATCH INVERSE-VARIANCE WEIGHTING: DEEP HETEROSCEDASTIC REGRESSION

## ABSTRACT

In model learning, when the training dataset on which the parameters are optimized and the testing dataset on which the model is evaluated are not sampled from identical distributions, we say that the datasets are misaligned. It is well-known that this misalignment can negatively impact model performance. A common source of misalignment is that the inputs are sampled from different distributions. Another source for this misalignment is that the label generating process used to create the training dataset is imperfect. In this work, we consider this setting and additionally assume that the label generating process is able to provide us with a quantity for the role of each label in the misalignment between the datasets, which we consider to be privileged information. Specifically, we consider the task of regression with labels corrupted by heteroscedastic noise and we assume that we have access to an estimate of the variance over each sample. We propose a general approach to include this privileged information in the loss function together with dataset statistics inferred from the mini-batch to mitigate the impact of the dataset misalignment. Subsequently, we propose a specific algorithm for the heteroscedastic regression case, called Batch Inverse-Variance weighting, which adapts inverse-variance weighting for linear regression to the case of neural network function approximation. We demonstrate that this approach achieves a significant improvement in network training performances compared to baselines when confronted with high, input-independent noise.

## 1 INTRODUCTION

In supervised learning, a central assumption is that the samples in the training dataset, used to train the model, and the samples in the testing set, used to evaluate the model, are sampled from identical distributions. Formally, for input $\mathbf{x}$ and label $y$, this assumption implies that $p_{\text{train}}(\mathbf{x}, y) = p_{\text{test}}(\mathbf{x}, y)$. This assumption can be decomposed as the product $p_{\text{train}}(\mathbf{x}) \cdot p_{\text{train}}(y|\mathbf{x}) = p_{\text{test}}(\mathbf{x}) \cdot p_{\text{test}}(y|\mathbf{x})$, which is true if two conditions are respected:

1. The features in both datasets are sampled from the same distribution: $p_{\text{train}}(\mathbf{x}) = p_{\text{test}}(\mathbf{x})$. When this is condition is violated, the training dataset is *not representative*.
2. The labels in both datasets are sampled from the same conditional distribution: $p_{\text{train}}(y|\mathbf{x}) = p_{\text{test}}(y|\mathbf{x})$. If this condition is violated, the training labels are *noisy*.

In practice, these assumptions are not always respected because gathering representative and precise data (including labels) can be arduous. In this case, the training and testing datasets are *misaligned*, and the performance of the deployed model may decrease since the training process did not actually optimize the model's parameters based on the correct data (Arpit et al., 2017; Kawaguchi et al., 2020). One possible reason for misalignment is that there is some uncertainty about the labels in the training set as a result of the labeling process. Since our objective is to optimize the performance of the model compared to ground truth labels, we should consider that the labels in test dataset have no uncertainty, even though it may be impossible to collect such a dataset in practice. As a result, $p_{\text{test}}(y|\mathbf{x})$ is sampled from a Dirac delta function, whereas $p_{\text{train}}(y|\mathbf{x})$ is not since it encapsulates the uncertainty in the labelling process, which leads to misalignment.

In this paper, we propose an algorithm for more efficient model training in the case where we have some information about the sample-wise misalignment. More specifically, we examine the case of

regression with a deep network where labels are corrupted by heteroscedastic noise. We assume that we have access at least an estimate of the variance of the distribution of the noise that corrupted each label, information that is available if the labels are being generated by some stochastic process that is capable of also jointly reporting uncertainty. We examine how the knowledge of the estimate of the label noise variance can be used to mitigate the effect of the noise on the learning process of a deep neural network. We refer to our method as Batch Inverse-Variance (BIV), which, inspired by information theory, performs a re-weighting using both the the sample-wise variance but also statistics over the entire mini-batch. BIV shows a strong empirical advantage over L2 loss as well as over a simple filtering of the samples based on a threshold over the variance.[1]

Our claimed contributions are threefold:

1. A definition of the problem of learning with information quantifying the misalignment between datasets for the case of heteroscedastic noisy labels in regression.

2. A general formulation of how to use the mini-batch to infer statistics of the dataset and incorporate this information in the loss function when training on neural networks.

3. We present Batch Inverse-Variance as an instantiation of this framework and show its usefulness when applied to regression tasks with labels corrupted by heteroscedastic noise.

**The outline of the paper is as follows:** In section 2, we describe the task of regression with heteroscedastic noisy labels and its parallels with learning with privileged information, and we explain the challenges of applying classical heteroscedastic regression methods to stochastic gradient descent. In section 3, we position our work among the existing literature on learning with noisy labels. In section 4, we present a general framework to incorporate information regarding dataset misalignment in the mini-batch loss. We introduce BIV within this framework to tackle heteroscedastic regression. In section 5, we describe the setup for the experiments we made to validate the benefits of using BIV, and we present and analyze the results in section 6.

## 2 BACKGROUND

### 2.1 HETEROSCEDASTIC NOISY LABELS IN REGRESSION

Here, we introduce how heteroscedastic noisy labels can be generated in regression and how the variance can be known. Consider an unlabelled dataset of inputs $\{\mathbf{x}_i\}$. To label it, one must apply to each input $\mathbf{x}_i$ an instance of a label generator which should provide its true label $y_i$. This label generator has access to some features $\mathbf{z}_i$ correlated to $\mathbf{x}_i$. We define $LG_j : \mathcal{Z} \to \mathbb{R}$ . When the labelling process is not exact and causes some noise on the label, the noisy label of $\mathbf{x}_i$ provided by $LG_j$ is defined as $\tilde{y}_{i,j}$. Noise on a measured or estimated value is often represented by a Gaussian distribution, based on the central limit theorem, as most noisy processes are the sum of several independent variables. Gaussian distributions are also mathematically practical, although they present some drawbacks as they can only represent unimodal and symmetric noise (Thrun et al., 2006). We model:

$$\tilde{y}_{i,j} = y_i + \delta_{y_{i,j}} \text{ with } \delta_{y_{i,j}} \sim N(0, \sigma_{i,j}^2) \tag{1}$$

$\sigma_{i,j}^2$ can be a function of $\mathbf{z}_i$ and $LG_j$, without any assumption on its dependence on one or the other. We finally assume that the label generator is able to provide an estimate of $\sigma_{i,j}^2$, therefore being redefined as $LG_j : \mathcal{Z} \to \mathbb{R} \times \mathbb{R}_{\geq 0}$. The training dataset is formed of triplets $(\mathbf{x}_i, \sigma_{i,j}^2, \tilde{y}_{i,j})$, renamed $(\mathbf{x}_k, \sigma_k^2, \tilde{y}_k)$ for triplet $k$ for simplicity. This setup describes many labelling processes, such as:

**Crowd-sourced labelling:** In the example case of age estimation from facial pictures, labellers Alice and Bob are given $\mathbf{z}_i = \mathbf{x}_i$ the picture of someone's face and are asked to estimate the age of that person. Age is harder to estimate for older people come (5 and 15 years of age are harder to confuse than 75 and 85) suggesting a correlation between $\sigma_{i,j}^2$ and $\mathbf{z}_i$. But Alice and Bob may also have been given different instructions regarding the precision needed, inducing a correlation between $\sigma_{i,j}^2$ and $LG_j$. Finally, there may be some additional interactions between $\mathbf{z}_i$ and $LG_j$, as for example Alice may know Charlie, recognize him on the picture and label his age with lower

---

[1]Our code is available in supplemental material and will be publicly released after the reviewing process.

uncertainty. Both labellers can provide an estimation of the uncertainty around their labels, for example with a plus-minus range which can be used as a proxy for standard deviation.

**Labelling from sensor readings, population studies, or simulations:** Imagine you want build a dataset of pictures $\mathbf{x}_i$ from a camera on the ground labelled with the position $y_i$ of a drone in the sky. To estimate the position of the drone at the moment the picture was taken, you could use state estimation algorithms based on the Bayes' filter (Thrun et al., 2006). These algorithms take as an input $\mathbf{z}_i$ the measurements of the drone's sensors, and provide a full posterior distribution over the state, sometimes under a Gaussian assumption for Kalman filters for example. The uncertainty depends, among others, on the precision of the sensors, the observability of a given state, the precision of the dynamic model, and the time since sensor signals were received. Similarly, studies based on population such as polling or pharmaceutical trials have quantified uncertainties based on the quantity and quality of their samples. It is also possible to train on simulators, as in climate sciences (Rasp et al., 2018) or in epidemiology (Alsdurf et al., 2020), and some of them provide their estimations' uncertainty based on the simulation procedure and the inclusion of real measurements in the model.

**Using predictions from a neural network in complex neural architectures:** In deep reinforcement learning for example, the critic network learns to predict a value from a state-action pair under the supervision of the heteroscedastic noisy output of a target network plus the reward (Mnih et al., 2015; Haarnoja et al., 2018). While the estimation of the uncertainty of the output of a neural network is not an easy task, it is an active field of research (Gal & Ghahramani, 2016; Peretroukhin et al., 2019). There, $\mathbf{z}_i$ is the state-action pair at the next step, and $LG_j$ the target network being updated over time. The prediction is a mix of aleatoric and epistemic uncertainties as defined by Kendall & Gal (2017) which are dependent on both $\mathbf{z}_i$ and $LG_j$.

We could not find any current dataset that provides such label uncertainty information for regression. However, as it is precious information, we argue that it should actually be provided when possible. In classification, Xie et al. (2016; 2020) took a step in this direction by providing a "confidence" score from 0 to 255 for each pixel in the KITTI-360 dataset .

## 2.2 LEARNING USING PRIVILEGED INFORMATION

Training with a dataset of triplets $(\mathbf{x}_i, \mathbf{x}_i^*, y_i)$, where $\mathbf{x}_i^*$ is only given at training time and not available at test time, fits in the framework of learning using privileged information (LUPI), defined in Vapnik & Vashist (2009) and mainly applied to SVMs. In most works in this field, this privileged information makes the task easier on a sample-to-sample basis. For example, object detection can be improved by adding segmentation masks (Feyereisl et al., 2014) or depth images (Hoffman et al., 2016). Another interpretation of LUPI is to use privileged information as a vector for knowledge transfer between a teacher and a student (Vapnik & Izmailov, 2015). Hernández-Lobato et al. (2014) and Lambert et al. (2018) have made a link between privileged information and uncertainty, using it to evaluate the confidence of the model for a training sample. The former applied the approach to Gaussian processes through latent noise, and the latter to neural networks through Gaussian dropout.

More formally, at training time the neural network has access to triplets $(\mathbf{x}_k, \mathbf{x}_k^*, y_k)$ where $\mathbf{x}_k$ is the input, $y_k$ its corresponding label, and $\mathbf{x}_k^*$ the additional information with respect to this sample.

The objective in LUPI is the same as classical supervised learning: train the network parameters $\theta$ so that, at test time, and without access to information $\mathbf{x}_i^*$, the expected loss is minimized, i.e.:

$$\theta^{opt} = \arg\min_{\theta} \mathbb{E}_{\{\mathbf{x}_i, y_i\} \in D_{\text{test}}} \left[ \mathcal{L}\left( f(\mathbf{x}_i, \theta), y_i \right) \right] \tag{2}$$

where $\mathcal{L}(f(\mathbf{x}_i, \theta), y_i)$ is the objective loss function based on the true label and on the network's prediction $f(\mathbf{x}_i, \theta)$, for example the L2 distance in the task of regression.

In our work, we have $\mathbf{x}_i^* = \sigma_i^2$. In contrast with the usual LUPI setting, $\mathbf{x}_i^*$ does not help the task on a sample-to-sample basis, but instead informs about the role of each sample on the misalignment between the datasets due to the noise in the labelling process. The objective, however, is the same: use this privileged information during training to minimize the expected loss at test time.

## 2.3 HETEROSCEDASTIC REGRESSION FOR LINEAR MODELS

The task of heteroscedastic linear regression, where the model is linear, is solved by optimizing a weighted mean square error (WMSE) with inverse-variance weights, which is the optimal solution as per the Gauss-Markov theorem (Shalizi, 2019):

$$\sum_{i=0}^{n} \frac{y_i - \mathbf{x_i} \cdot \beta}{\sigma_i^2} \tag{3}$$

where $\beta$ is the vector of parameters used as linear coefficients. This is also the solution to maximum likelihood estimation for $\beta$ (Fisher, 1957).

While the solution to such an optimization is known for linear regression ($\beta^* = (\mathbf{x}^T \mathbf{w} \mathbf{x}^{-1}) \mathbf{x}^T \mathbf{w} \mathbf{y}$), several problems appear when attempting to adapt it to gradient-based methods on neural networks, such as stochastic gradient descent: (1) the learning rate in gradient-based methods impacts the optimization process in multiple ways and should be controllable by the practitioner regardless of the amount of noise in the samples to prevent very small or large gradients from destabilizing the learning process (2) similarly, near ground-truth samples should not have a disproportionate learning rate with respect to the others, as they risk to cause overfitting.

In our work, we propose a method to apply such weights to neural networks while addressing these issues.

## 3 RELATED WORK

Noise on labels amounts to a loss of information. When the noise is significant enough, it leads to overfitting and lower model performance (Liu & Castagna, 1999; Zhang et al., 2017). This effect is more prevalent in small data settings (Van Horn et al., 2015). Four possible strategies exist in the literature to tackle this problem: detection, correction, robustness, or re-weighting. **Detection** consists of identifying noisy labels and ignoring them in the learning process. These methods are often based on the observation that neural networks first fit on consistent, non-noisy data (Arpit et al., 2017), thus converging to a higher loss on the noisy samples (Reed et al., 2015; Shen & Sanghavi, 2019). Other methods use several neural networks to co-teach each other (Han et al., 2018; Yu et al., 2019) or dropout to estimate the consistency of the data (Reed et al., 2015). However, in the case of imbalanced training datasets, higher loss can also be the signature of a non-noisy but rare sample. Cao et al. (2020) address this ambiguity by regularizing different regions of the input space differently. **Correction** strategies go further: once noise is detected, the noisy labels are changed to probability distributions. Such an operation requires a noise model. Goldberger & Ben-Reuven (2017); Kremer et al. (2018); Ma et al. (2018); Tanno et al. (2019); Yi & Wu (2019) learn it jointly with the parameters, assuming a correlation between the noise and the input, the labels, or both. **Robust loss functions** are less sensitive to noise. Liu & Castagna (1999) proposed to avoid overfitting due to noise by ignoring samples during the training when the prediction error is reasonable. Natarajan et al. (2013) compute the loss assuming knowledge of example-independant mislabelling probabilities in binary classification, and then optimize these hyperparameters with cross-validation. More recent works are based on reverse cross-entropy (Wang et al., 2019) or curriculum loss (Lyu & Tsang, 2020). Others leverage a distillate of the information gathered from a subset of clean labels to guide training with noisy labels (Li et al., 2017). **Re-weighting** the samples is another efficient method for mitigating noise in datasets. Liu & Tao (2016) estimate the effective label probabilities as well as noise rates for a given input and use these estimates to weigh the samples using importance sampling. Shu et al. (2019) go one step further by learning the weighting function through a meta-learning method. Jenni & Favaro (2018) control overfitting by adjusting sample weights in the training and validation mini-batches, increasing robustness to overfitting on noisy labels.

While most works that address noisy labels consider classification tasks (Song et al., 2020), only some of these strategies can be generalized to regression. Heteroscedastic regression occurs when each label's noise is sampled from a different distribution. Nix & Weigend (1994) tackle this problem in neural networks by jointly training a variance estimator based on the maximum likelihood of an underlying Gaussian model. Kendall & Gal (2017) use the same idea to estimate the aleatoric (input-dependant) uncertainty of the network's prediction, while using dropout as a Bayesian ap-

proximation for the epistemic uncertainty (due to the learning process) as in (Gal & Ghahramani, 2016).

Our method tackles heteroscedastic regression in neural networks using a re-weighting approach. The main distinction between our work and most of the related literature is that, while we do not require that the noise variance is a function of the input or of the label, we do assume that we have access to the noise variance, or at least an estimate of it. In addition, we do not seek to regress the variance of the model's prediction. This is significant compared to the previous works in both regression and classification as it changes the loss function and removes the need for a regularizer for the variance prediction.

## 4 INCORPORATING PRIVILEGED INFORMATION IN THE LOSS FUNCTION

In this section, we first present a general operator to incorporate privileged information and infer dataset statistics on the loss computed at the mini-batch level. Then, we describe our solution, BIV, an instance of this operator for heteroscedastic regression. Finally, we introduce a more basic filtering function which we will use as a baseline in our experiments.

### 4.1 GENERAL OPERATOR FOR TRAINING ON THE MINI-BATCH LEVEL

The value of privileged information about misalignment between the training and testing datasets is often higher when combined with statistics about the datasets. For example, in the case of noisy labels, the uncertainty on each label is relevant when compared to the information carried by the other samples, similarly to (3).

We propose to both incorporate the privileged information and infer dataset statistics during the training of neural networks over a mini-batch, as opposed to the individual samples. There are two main advantages in doing so. First, if the mini-batch samples are independently and identically sampled from the dataset, they can be used to infer some statistics of the whole dataset. Working on the mini-batch level allows us to use such an approach without any pre-processing step over the whole dataset, which is important for many tasks such as continuous learning. Second, by focusing on the single step of computing the loss, this approach does not interfere with any of the variety of other methods used to optimize the learning process, such as regularization, batch normalization, annealing learning rates, etc.

In general, this approach can be expressed by defining an operator $\mathcal{G}$ applied on the objective loss function. For a mini-batch $D_i$ of $K$ sample triplets, we define the loss as:

$$\mathcal{L}_{\text{batch}}(D_i, \theta) = \mathcal{G}\left(\mathbf{x}_{1:K}, \mathbf{x}^*_{1:K}, \mathbf{y}_{1:K}, \mathcal{L}\left(\cdot, \cdot\right)\right) \tag{4}$$

Note that without any privileged information, operator $\mathcal{G}$ is usually the unweighted average of the loss computed over each sample of the batch, which is equivalent to empirical risk minimization.

### 4.2 BATCH INVERSE-VARIANCE WEIGHTING FOR HETEROSCEDASTIC NOISY LABELS

To tackle the problem of heteroscedastic regression in neural networks, we follow the intuition of equation (3) and describe $\mathcal{G}$ as a weighted average with weights $w_k = 1/\left(\sigma_k^2 + \epsilon\right)$. We introduce the Batch Inverse-Variance (BIV) loss function:

$$\mathcal{L}_{\text{batch}}(D_i, \theta) = \left(\sum_{k=0}^{K} \frac{1}{\sigma_k^2 + \epsilon}\right)^{-1} \sum_{k=0}^{K} \frac{\mathcal{L}\left(f(\mathbf{x}_k, \theta), \tilde{y}_k\right)}{\sigma_k^2 + \epsilon} \tag{5}$$

Here, the inverse of the sum has two major roles. It is a normalization constant for the mini-batch, allowing to keep a consistency in the learning rate. Note that consistency is verified as, when $\sigma_k^2$ is identical for each sample, this formulation leads to empirical risk minimization.

The hyper-parameter $\epsilon$ is effectively a lower bound on the variance. This allows us to incorporate samples with ground-truth labels without completely ignoring the other samples. The choice of $\epsilon$ is a trade-off between regulating the weights of near-ground-truth labels and using BIV at its full capacity. We found that it can be set between $0.01$ and $0.1$ and use $\epsilon = 0.1$. More details on $\epsilon$ can be found in appendix B.1.

These two elements, added to the advantages of computing the loss function in the mini-batch only, allow us to overcome the challenges related to inverse variance weighting applied to gradient descent, as described in section 2.3.

### 4.3 CUTOFF: FILTERING HETEROSCEDASTIC NOISY LABELS

As we do not assume that there is a correlation between $\mathbf{x}_k$ and $\sigma_k^2$, most correction and re-weighting algorithms as presented in section 3 are not applicable. Most robust loss function are specifically designed for classification problems. We thus compare BIV to a detection and rejection strategy.

In heteroscedastic regression, an important difference from classification with noisy labels is that all labels are corrupted, albeit not at the same scale. Defining which labels to ignore is therefore a matter of putting a threshold on the variance. As we have access to this information, strategies such as the ones used in section 3 are not necessary. Instead, we simply use an inverse Heaviside step function as a weight in the loss function:

$$w_k = \mathbf{1}_{\sigma_k^2 < C} \tag{6}$$

where the threshold $C$ is a hyper-parameter. Similarly to equation (5), we normalize the loss in the mini-batch by the sum of the weights, equal here to the number of samples considered as valid. As this filtering is equivalent to cutting off a part of the dataset, we refer to this method as 'Cutoff', and consider it to be a relevant baseline to compare BIV against.

## 5 EXPERIMENTAL SETUP

To test the validity of the BIV loss (5) approach, we compared its performance with the classical L2 loss as well as cutoff loss (6) on two datasets. We refer to ground-truth (GT) labels when training with L2 on noise-less data as the best performance that could be achieved on this dataset.

Unfortunately, we did not find any existing dataset for regression where label uncertainty is associated to the samples. We therefore used two UCI datasets (Dua & Graff, 2017) for regression cases, and artificially added noise to them. UTKFace Aligned&Cropped (Song & Zhang, 2017) (UTKF), is a dataset for image-based age prediction. In the Bike Sharing dataset (Fanaee-T & Gama, 2013), the task is to predict the number of bicycles rented in Washington D.C., from structured data containing the date, hour, and weather conditions. For UTKF, a convolutional neural network was used to predict the age, while a simple multi-layer perceptron was used for BikeSharing. More details about the datasets and models can be found in appendix A.

### 5.1 NOISE GENERATION

To produce the datasets $\{\mathbf{x}_k, \sigma_k^2, \tilde{y}_k\}$ with noise as described in section 2.1, we use a two-step process which does not assume any correlation between the noise and the state.

1. the noise variance $\sigma_k^2$ is sampled from a distribution $P(\sigma^2)$ which only has support for $\sigma^2 \geq 0$
2. $\tilde{y}_k$ is sampled from a normal distribution $\mathcal{N}(y_k, \sigma_k^2)$.

$P(\sigma^2)$ has a strong effect on the impact of BIV or Cutoff. For example, if it is a Dirac delta and all variances are the same, BIV becomes L2. We evaluate BIV on three different types of $P(\sigma^2)$. The average noise variance $\mu_P$ was chosen empirically so that the lowest test loss achieved by L2 is doubled compared to the ground-truth label case: $\mu_P = 2000$ for UTKF and 20000 for BikeSharing.

**Uniform distribution** The uniform distribution is characterized by its bounds $a, b$. Its expected value $\mu_P$ is the average of its bounds, and its variance $V = (b-a)^2/12$. As $P$ only has support for $\sigma^2 \geq 0$, the maximum variance $V_{\max}$ is when $a = 0$ and $b = 2\mu_P$. While such a distribution is not realistic, it is simple conceptually and allows for interesting insights.

**"Binary uniform"** A more realistic distribution, which can also help us understand the effects of BIV, is when the data is generated from two regimes: low and high noise. We call the "binary uniform" distribution a mixture of two uniform distributions balanced by parameter $p$.

With probability $p$, the label is in a low noise regime: $\sigma^2 \sim U(0,1)$, with expected value $\mu_l = 0.5$. With probability $1-p$, the label is in a high noise regime: $\sigma^2 \sim U(a_h, b_h)$. $a_h$ and $b_h$ are chosen such that the average is $\mu_h$ and the variance of the high-noise distribution is determined by $V_h \in [0, V_{\max}]$. Note that, if we want a given expected value of the whole distribution $\mu_P$, the value of $\mu_h$ changes depending on $p$: $\mu_h = (\mu_P - p\mu_l)/(1 - p)$

Therefore, the high-noise expected value $\mu_h$ of the noise variance $\sigma^2$ in a distribution with high $p$ will be higher than the one for a low $p$, for the same value of $\mu_P$. In other words, a higher $p$ means more chance to be in the low-noise regime, but the high-noise regime is noisier.

**Gamma distributions**  While the mixture of 2 uniform distributions ensures support in the low noise region, it is not continuous. We therefore also propose to use a Gamma distribution with shape parameter $\alpha$. If we want to control the expected value $\mu_P$, we adjust $\beta = \alpha/\mu_P$.

For a fixed expected value $\mu_P$, lower $\alpha$ and $\beta$ mean that there is a stronger support towards low variance noise, but the tail of the distribution spreads longer on the high-noise size. In other words, a lower $\alpha$ means more chances to have low-noise samples, but when they are noisy, the variance is higher. When $\alpha \leq 1$, the highest support is at $\sigma^2 = 0$.

## 5.2 EVALUATING THE PERFORMANCE OF THE MODEL

The objective of BIV is to improve the performance at predicting the true label $y_i$, as mentioned in equation (2). While a non-noisy test dataset may not be available in a real application, we aimed here at determining if BIV performs better than L2, and therefore measured the performance of the network using ground-truth test data.

## 6 EXPERIMENTAL RESULTS AND ANALYSIS

### 6.1 FOR L2 LOSS, MEAN VARIANCE IS ALL THAT MATTERS

Before looking at the results for BIV, we share an interesting insight for L2 loss with noisy labels which helps simplifying the analysis of the results. Under the unbiased, heteroscedastic Gaussian-based noise model presented in section 5.1, the only parameter of distribution $P(\sigma)$ that mattered to describe the performance of the L2 loss is its average $\mu_P$, which is also the variance of the overall noise distribution. Independently of the distribution type, and the values of $V$, $p$ and $V_h$, or $\alpha$, as long as $\mu_P$ is equal, the L2 loss trained neural networks had the same performance. This is shown in Figure 1. For the sake of clarity, all the curves in this section were smoothed using moving average with a 35 steps window, and the shaded area represents the standard deviation over 10 runs.

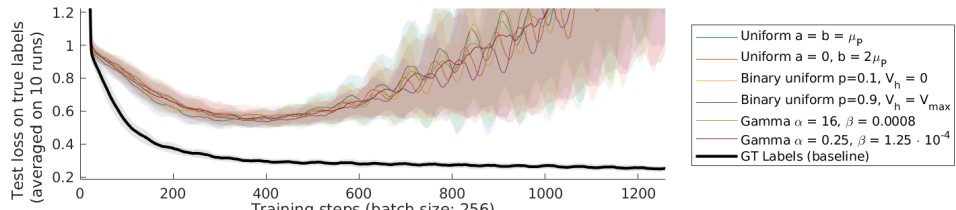

Figure 1: Performance of the neural network on UTKF trained with L2 loss, for different $P(\sigma^2)$ with constant $\mu_P = 2000$. No matter the distribution type or parameters, the performance is similar.

### 6.2 HIGH AND LOW VARIANCE NOISE REGIMES: BIV ACTS AS A FILTER

With the binary uniform distribution, the noise is split in two regimes, with high or low variances. In this case, our results show that BIV performs better than L2, and actually similarly to the cutoff loss presented in section 4.3 with a threshold $C = 1$.

Figure 2 compares the test losses on UTKF with different values of $p$ for $V_h = 0$. While the L2 curves are strongly impacted by the noise, both the BIV and cutoff losses lead to better and very similar performances for a given $p$. When $p = 0.3$, there are not a lot of information that can be

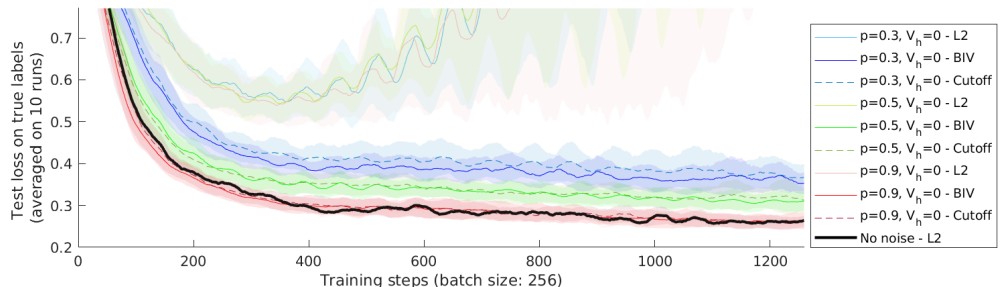

Figure 2: Comparison between BIV, Cutoff and L2 losses for binary uniform distributions of variance with different $p$s for $\mu_P = 2000$ and $V_h = 0$ on UTKF.

used, and the performance is still impacted by the noise. When $p = 0.9$, there is nearly as much near-ground-truth data as in the noiseless case, and the performance is comparable.

In the case of binary uniform distributions, BIV is acting as a filter, cutting off labels which are too noisy to contain any useful information.

### 6.3 CONTINUOUS, DECREASING NOISE VARIANCE DISTRIBUTION: THE ADVANTAGE OF BIV

On Gamma distributions, there is no clear threshold to define which information to use. When $\alpha \leq 1$, BIV shows a strong advantage compared to both L2 and cutoff. Figure 3 shows the results in both the BikeSharing and the UTKF datasets for Gamma distributions with $\alpha = 1$.

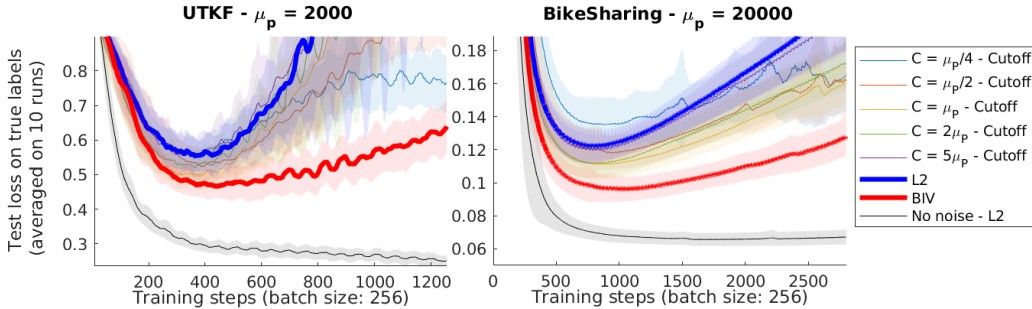

Figure 3: Comparison between the performances of BIV, L2, and different cutoff values on both datasets where the noise variance follows a Gamma distribution with $\alpha = 1$.

In both datasets, when the cutoff parameter $C$ is too low ($\mu_P/4$ and $\mu_P/2$), there is not enough data to train the model. When $C$ is too high ($2\mu_P$ and $5\mu_P$), the data is too noisy and the curves go close to the original L2 loss. Even at the best case ($C = \mu_P$), cutoff is not better than BIV. This is because, in contrast to cutoff, BIV is able to extract some information from noisier samples while avoiding to overfit on them.

In Table 1, we present the lowest value of the test loss curves for the different methods with other $\alpha$ parameters for the Gamma distributions over both datasets. BIV consistently leads to the best performances, regardless of $P(\sigma^2)$. The plots showing these runs can be found in the appendix B.3.1. BIV is less sensitive to hyperparameters than cutoff, as it avoids the need to choose the right cutoff parameter for each distribution $P(\sigma^2)$. BIV's own hyperparameter $\epsilon$ can be set between 0.01 and 0.1 for any dataset with a normalized output, as shown in appendix B.1, and be ready to use. As $\epsilon$ can be seen as a minimal variance, scaling it for other label distributions is straightforward: it suffices to multiply it by the variance of the label distribution.

The benefit of BIV over L2 is clearly higher when $\alpha$ is lower. This is due to an increase in the support for low-variance noise in $P(\sigma^2)$. The more BIV can count on low-noise elements and differentiate them from high noise ones, the better it can perform. This is consistent with results from section

Table 1: Lowest test loss for different $\alpha$ on two datasets, for BIV, L2 and several cutoff losses. The test loss with standard deviation is computed as the average over 10 runs. In every case, BIV loss led to the lowest value. The best $C$ value differs based on $\alpha$.

|  | $\alpha = 1$ | | $\alpha = 0.5$ | | $\alpha = 0.25$ | |
|---|---|---|---|---|---|---|
|  | UTKF | Bike | UTKF | Bike | UTKF | Bike |
| $C = \mu_P/20$ | 0.79±.08 | 0.327±.056 | 0.48±.04 | 0.125±.010 | 0.38 ±.05 | 0.092±.008 |
| $C = \mu_P/4$ | 0.55±.04 | 0.135±.016 | 0.45±.04 | 0.097±.006 | 0.39±.03 | 0.085±.006 |
| $C = \mu_P$ | 0.50±.04 | 0.111±.009 | 0.48±.04 | 0.097±.008 | 0.43±.03 | 0.088±.006 |
| $C = 5\mu_P$ | 0.55±.06 | 0.120±.009 | 0.54±.05 | 0.111±.012 | 0.51±.04 | 0.107±.009 |
| L2 | 0.56±.05 | 0.122±.010 | 0.56±.05 | 0.116±.011 | 0.55±.05 | 0.119 ±.012 |
| BIV (ours) | **0.47**±.03 | **0.096**±.007 | **0.41**±.03 | **0.084**±.006 | **0.34**±.02 | **0.079**±.006 |
| GT labels | 0.25±.02 | 0.066±.004 | 0.25±.02 | 0.066±.004 | 0.25±.02 | 0.066±.004 |

6.2, and with other experiments we have run. For example, when $\alpha > 1$, the highest support of $P$ is not at $\sigma^2 = 0$. BIV was less able to improve the performance compared to L2.

We also ran the experiment with uniform distributions: the performance is better when variance $V$ is closer to $V_{max}$ (and $a$ to 0). But even when $V = V_{max}$, as there is less support in low noise variance than for Gamma distributions with $\alpha \leq 1$, the improvement is less important.

In all cases, BIV was performing consistently better than L2 and at least better than cutoff in all the experiments we ran. More details on these results can be found in appendix B.2.

## 6.4 ROBUSTNESS

We identified two elements that could impact the performances of BIV: the size of the mini-batches and the accuracy of the noise variance estimation. We tested the robustness of BIV when these factors are different than during our experiments.

**Size of the mini-batches**   In equation 5, each weight is normalized based on the assumption that the distribution of noise variances in the mini-batch is representative of the one in the whole training dataset. While this is less the case with smaller mini-batches, our results show that BIV still performs very well in these cases, as presented in section B.4.1.

**Noisy variances**   Because the noise variance $\sigma_i^2$ is often estimated, the method needs to be robust to errors in $\sigma_i^2$'s. A model for the noise of $\sigma_i^2$ can be a Gaussian for which the variance is proportional to $\sigma_i^2$. In this case, results show that the effect of moderate to high levels of noise on BIV is not significant. More details can be seen in section B.4.2

## 7 CONCLUSION

We have proposed a mini-batch based approach to incorporate in the loss function privileged information which quantifies the participation of each sample to the misalignment between the training and testing datasets. We described how such a setup can occur in the case of regression with heteroscedastic noisy labels. To tackle this problem, we introduced BIV, a method to apply inverse-variance weights in stochastic gradient descent. BIV is able to extract more information from the noisy dataset than L2 loss or threshold-based filtering approaches, and consistently outperforms them on both structured and unstructured datasets. BIV can improve the performance of supervised learning in many heteroscedastic regression scenarios, where the label is generated by a process such as crowd-labelling, sensor-based state estimation, simulation, or complex neural architectures. More generally, the framework for including privileged information quantifying the datasets misalignment in the loss function on a mini-batch level could be used to account for other types of misalignment, such as under-represented hidden features or correlation between samples.

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

## A    APPENDIX - DATASETS AND NEURAL NETWORKS

### A.1    UTKFACE

#### A.1.1    DATASET DESCRIPTION

The UTKFace Aligned&Cropped dataset (Song & Zhang, 2017) consists of 20,000 pictures of faces labelled with their age, ranging from 0 to 116 years. We use it in a regression setting: the network must predict the age of a person given the photo of their face. Unless described otherwise, 16,000 images were used for training, and 4,000 for testing.

Some images are in black and white and some are in color. The pixel dimension of each image is 200x200.

Both the pixels and the labels were normalized before the training, so that their mean is 0 and standard deviation is 1 over the whole dataset. The noise variances were correspondingly scaled, as well as the cutoff threshold if applicable.

#### A.1.2    NEURAL NETWORK AND TRAINING HYPER-PARAMETERS

The model that we used was a Resnet-18 (He et al., 2015), not pretrained. It was trained with an Adam optimizer (Kingma & Ba, 2017), a learning rate of 0.001 over 20 epochs. A batch size of 256 was used in order to ensure the best performance for the L2 method with noisy labels as well as to reduce the time necessary to the training process.

### A.2    BIKE SHARING DATASET

#### A.2.1    DATASET DESCRIPTION

The Bike Sharing Dataset (Fanaee-T & Gama, 2013) consists of 17,379 samples of structured data. It contains, for nearly each hour of the years 2011 and 2012, the date, season, year, month, hour, day of the week, a boolean for it being a holiday, a boolean for it being a working day, the weather situation on a scale of 4 (1: clear and beautiful, 4: stormy or snowy), the temperature, the feeling temperature, the humidity, and the wind speed, in the city of Washington DC. It also contains the number of casual, registered, and total bike renters for each hour as recorded on the Capital Bikeshare system.

We use it in a regression setting: the network must predict the total number of bike renters given the time and weather information. Unless described otherwise, 7,000 samples were used for training, and 3,379 for testing. We used less samples than available for training because the low-data situation, noise has a stronger effect on the performance. The minimal test loss achieved with 7000 noiseless samples was very close to the one with 14000 samples, hinting that the additional samples did not give a lot of additional information.

We applied some pre-processing on the data to make it easier for the network to learn. First, the date was normalized from a scale between day 1 to day 730 to a scale between 0 and $4\pi$. Then, we provided the network with the cosine and the sine of this number. This allowed to have the same representation for the same days of the year, while having the same distance between any two consecutive days, keeping the cyclic nature of a year. A similar idea was applied to hours, normalized from 0 to $2\pi$ instead of 0 to 24, and with the cosine and sine given to the network. The day of the week, being a category, was given as a one-hot vector of dimension 7. We also removed the season and the month as it was redundant information with the date.

Overall, the number of features was 19:

    1  Year

  2-4  Date (sine and cos)

4-5 Hour (sine and cos)

6 to 12 Days of the week (one-hot vector)

13 Holiday boolean

14 Working day boolean

15 Weather situation

16 Temperature

17 Felt temperature

18 Humidity

19 Wind speed

We observed that the network was learning significantly faster and better provided with this format for the data.

Both the features and the labels were normalized before the training, so that their mean is 0 and standard deviation is 1 over the whole dataset. The noise variances were correspondingly scaled, as well as the cutoff threshold if applicable.

### A.2.2 NEURAL NETWORK AND TRAINING HYPER-PARAMETERS

The model that we used was a multi-layer perceptron with 4 hidden layers, the first one with 100 neurons, then 50, 20, and 10. The activation function was ReLU. We did not use any additional technique such as batch normalization as it did not improve the performances.

The model was trained over 100 epochs on mini-batches of size 256 for similar reasons than explained in section A.1.2, using the Adam optimizer with learning rate 0.001.

## B APPENDIX - ADDITIONAL EXPERIMENTS

### B.1 THE INFLUENCE OF $\epsilon$

In this section, we provide experimental results justifying our recommendation of the range of $[10^{-2}; 10^{-1}]$ for $\epsilon$. In the experiments presented in this article, we have used $\epsilon = 10^{-1}$. $\epsilon$ must be chosen as part of a trade-off between mitigating the effect of BIV with near ground-truth labels while keeping its effect with noisy labels. To better understand these results, it is important to remember that $\epsilon$ is added to the variance that is used in the loss function. When the labels are normalized - which is the case in our work -, the noise and its variance for each label is normalized too. The value we recommend for $\epsilon$ should therefore be valid for any normalized set of labels.

### B.1.1 $\epsilon$ FOR BIV WITH NEAR GROUND-TRUTH LABELS

One of the main problems BIV induces is in the presence of ground-truth or near ground-truth (NGT) labels. In this case, a near-zero variance can induce an very strong weight, effectively reduce the mini-batch to this single sample, and thus ignore the other potentially valid samples.

In this case, hyperparameter $\epsilon$ is key, as it allows to set a maximal weight and has a stronger relative effect on the NGT labels than on noisier ones. We tested on the UTKF dataset with a uniform distribution of variance from $a = 0$ to $b = 1$, which is effectively very little noise for the L2 loss but, by allowing NGT labels, can already showcase the influence of $\epsilon$ on the BIV performance. The resulting graph can be seen in figure 4.

It is clear that a very small $\epsilon$, such as $10^{-6}$ or $10^{-5}$, leads to a significant loss of performance. However, over $10^{-2}$, the performance is as good as the L2 loss. Note that it is not surprising when $\epsilon$ is high such as $10^2$, as in this case the weights are very close to similar and BIV is effectively the same as L2.

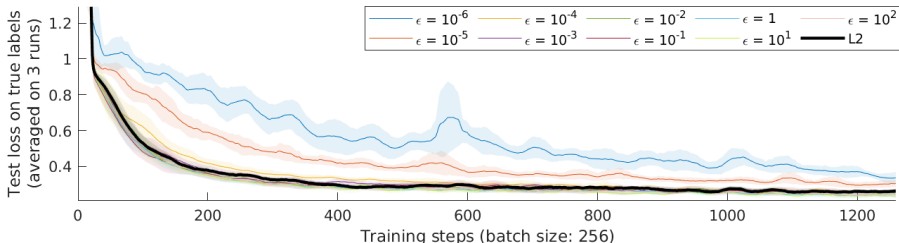

Figure 4: Impact of $\epsilon$ when training with NGT labels using BIV loss on UTKF dataset. The variance was sampled through a uniform distribution bound between 0 and 1. Very low $\epsilon$ shows a drastic loss of performance compared to L2 loss, as NGT labels monopolize the weights in the mini-batch.

### B.1.2 A HIGH $\epsilon$ REDUCES THE ADVANTAGES OF BIV ON NOISY LABELS

As discussed in the previous section, a higher value of $\epsilon$ makes the weights more similar for each of samples, and therefore reduces the effect of BIV. We tested BIV with different values of $\epsilon$ with a binary distribution with $p = 0.5$ and $\mu_P = 2000$ on UTKF. In this setup, BIV should have enough data to train correctly when filtering out the labels in the noisy regime, as shown in section 6.2. The results are shown in figure 5.

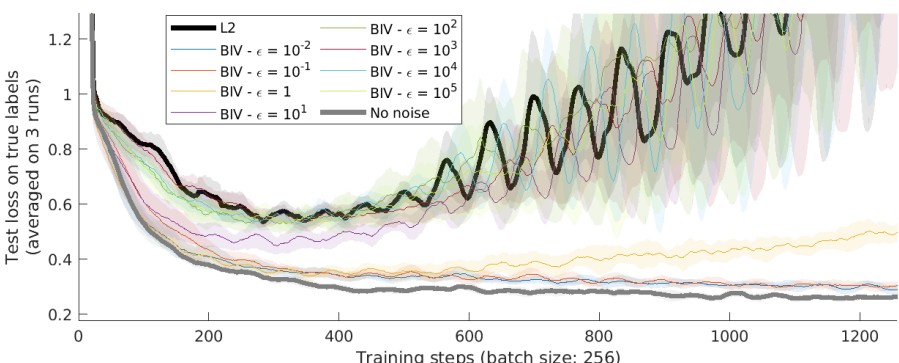

Figure 5: Impact of $\epsilon$ when training with highly noisy labels using BIV loss on UTKF dataset. The variance was sampled through a binary uniform distribution with $p = 0.5$, $\mu_P = 2000$, and $V_h = 0$. Very high $\epsilon$ shows a loss of performance as BIV approaches the L2 results.

As expected, when $\epsilon$ is very high, the results are very similar to L2. The first effects of BIV can be seen when $\epsilon = 10$, but until $\epsilon = 1$ it is still not optimal. From $\epsilon = 0.1$, the algorithm is correctly filtering the data.

Considering the results from sections B.1.1 and B.1.2, we consider that $\epsilon$ should be between $10^{-2}$ and $10^{-1}$ to optimize the balance between regulating the importance of near ground-truth labels and benefiting from the effect of BIV.

### B.2 BIV ON DIFFERENT DISTRIBUTIONS

### B.2.1 UNIFORM DISTRIBUTIONS

We present in figure 6 the results of the experiment with uniform distributions in more details.

As explained in section 6.3, we observe that BIV and L2 have the same performances when $V = 0$ (and $a = b = \mu_P$). This is to be expected, as all samples have the exact same noise variance and thus the same weights. When $V = V_{max}$ ($a = 0$ and $b = 2\mu_P$), BIV has an advantage, as it is able to differentiate the samples and use the support of low-noise labels. When $V = V_{max}/2$ ($a = 0.293\mu_P$ and $b = 1.707\mu_P$), the difference between the samples is less important, and BIV only does a bit

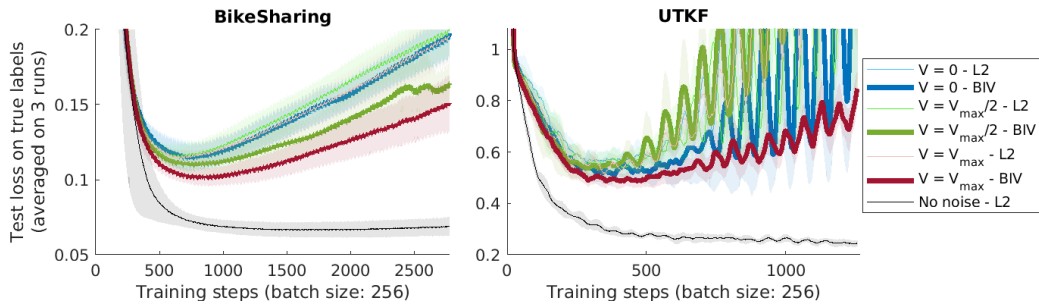

Figure 6: Test loss for L2 and BIV learning on uniform with different variances $V$.

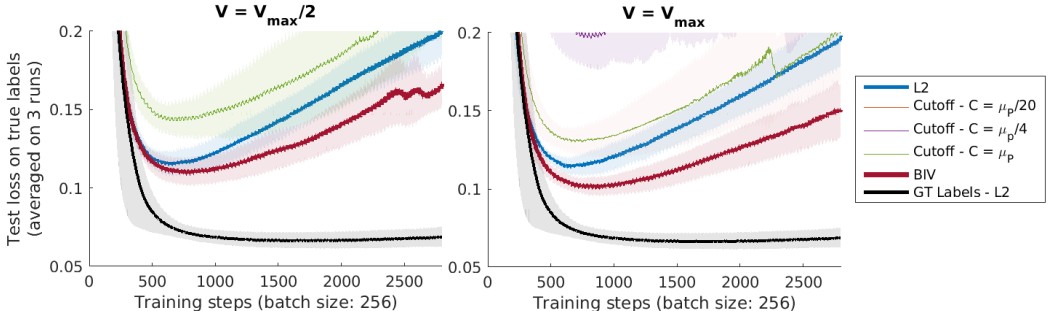

Figure 7: On BikeSharing with $\mu_P = 20000$, using cutoff is not helpful in the uniform setting. This is due to the significant loss of information induced by such a strategy.

better than L2 on BikeSharing. On UTKF, the process has more variability and it is difficult to detect this effect.

In all cases, the benefit from using BIV is less important than with Gamma distributions with $\alpha \leq 1$, where the support on low-noise samples is higher.

In this setting, we also show as shown in figure 7 that cutting off the noisy data is not a good strategy, as it always performs worse than L2.

### B.3 BIV on Gamma distributions

#### B.3.1 $\alpha \leq 1$

As described in section 6.3, the smaller $\alpha$, the better the performance of BIV and cutoffs. We show in figures 8 and 9 the curves that led to the numbers in Table 1. BIV consistently outperforms the other methods. The performance of cutoff methods strongly depends on $C$, and the best value of $C$ is not the same for every distribution $P$.

#### B.3.2 $\alpha > 1$

When $\alpha > 1$, the highest support of $P(\sigma^2)$ shifts towards $\mu_P$. This makes the samples less distinguishable for BIV and therefore the benefits of using it are reduced. This is shown in Figure 10 on UTKF.

### B.4 Robustness of BIV

#### B.4.1 Size of the mini-batches

In equation (5), the normalization constant is computed from the samples in the mini-batch. If the distribution of the noise variances in mini-batch is representative of the whole training dataset, the relative weight given to each sample with respect to the others is the same than if the normalization was made over the whole dataset. The larger the mini-batch, the more representative it is. In our

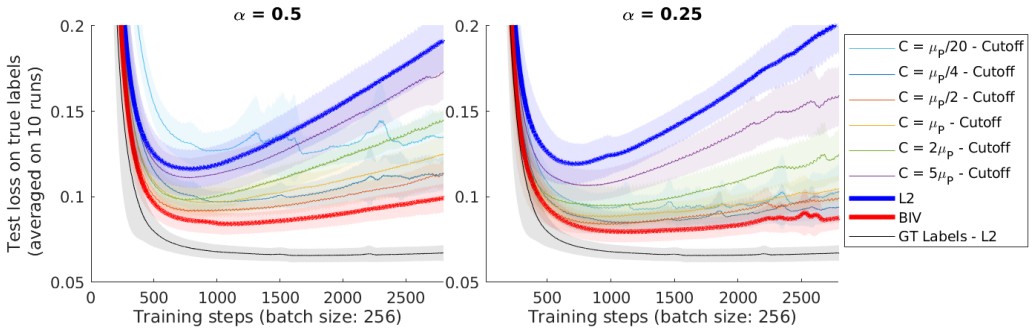

Figure 8: Test loss on the BikeSharing dataset, with $\alpha \leq 1$

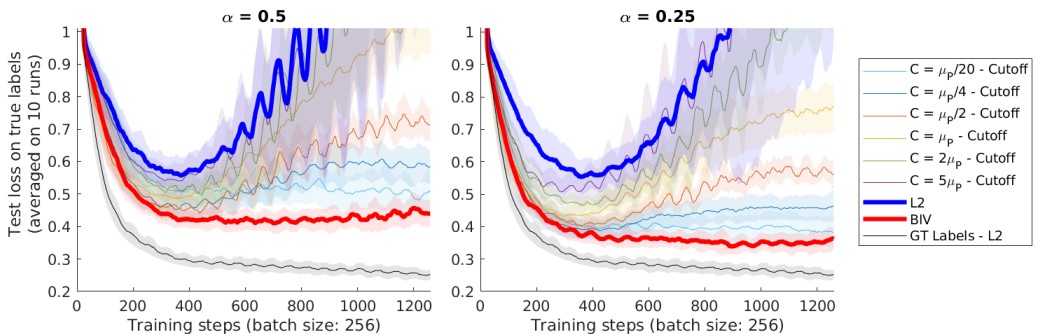

Figure 9: Test loss on the UTKF dataset, with $\alpha \leq 1$

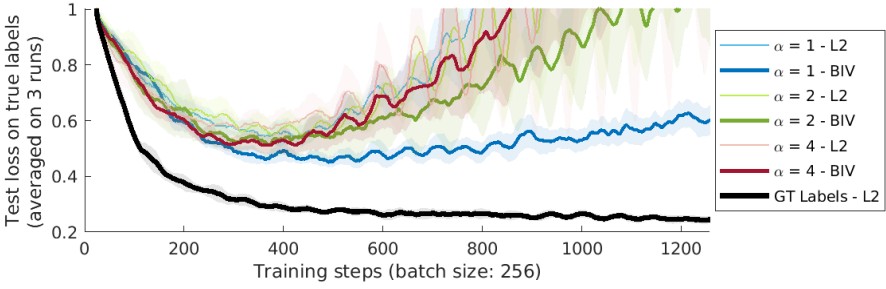

Figure 10: Test loss on the UTKF dataset for L2 and BIV learning on Gamma function with $\alpha \geq 1$.

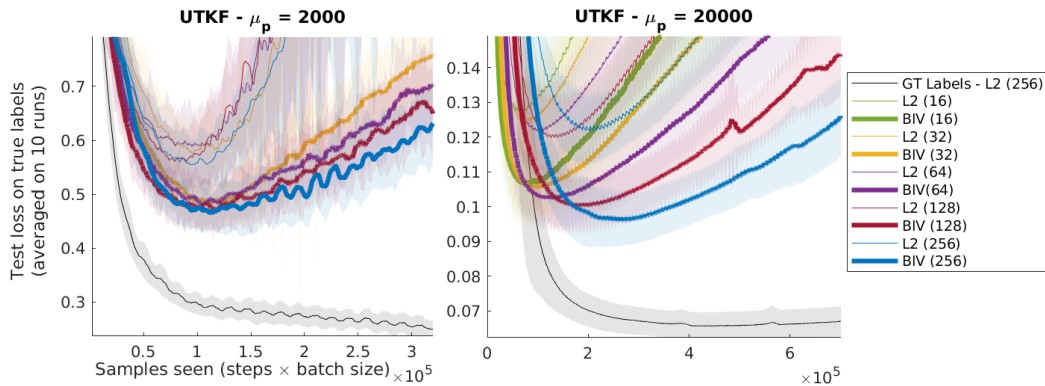

Figure 11: BIV with different batch sizes in both UTKF and BikeSharing datasets.

experiments, we used a size of 256, which is arguably high. We tested our algorithm with lower batch sizes, from 16 to 128, to see if it was a critical factor in the performances.

The results are presented in figure 11. In UTKF, the batch size does not make any significant difference in the performance with respect to the amount of samples seen, except for a slightly steeper overfitting once the best loss has been achieved. In BikeSharing, a smaller batch size makes the training faster with respect to the amount of samples, but with a higher minimal loss, for both L2 and BIV. While a larger batch size leads to a lower loss function, the effect of BIV compared to the corresponding L2 curve is not compromised by smaller batch-sizes.

Two main factors may explain this robustness. First, a mini-batch of size 16 seems to be already representative enough of the whole dataset for the purpose of normalization. Second, the fact that the mini-batches are populated differently at every epoch improves the robustness as a sample who would have been in a non-representative batch at one epoch may not be at another epoch. In any case, the size of the mini-batch is not a critical factor for BIV.

### B.4.2 NOISY VARIANCE ESTIMATION

In many scenarios, the variance $\sigma^2$ from which the noise was sampled is estimated, or inferred from a proxy, and therefore prone to be noisy itself. We tested the robustness of our method to such variance noise. In this experimental setup, the value given to the BIV algorithm is disturbed by noise $\delta_{\sigma^2}$. We modelled this noise on $\sigma_i^2$ to be sampled from a normal distribution whose standard deviation is proportional to $\sigma_i^2$ with a coefficient of variance disturbance $D_v$:

$$\delta_{\sigma_i^2} \sim \mathcal{N}(0, D_v \sigma_i^2/9) \tag{7}$$

Dividing $\sigma_i^2$ by 9 allows to scale $D_v$ so that, when $D_v = 1$, $\delta_{\sigma^2} < -\sigma_i^2$ is at 3 standard deviations from the mean of the distribution.

We then compute the noisy variance, which needs to be positive, as $\tilde{\sigma}_i^2 = \left|\sigma_i^2 + \delta_{\sigma_i^2}\right|$. The noise is therefore biased, but when $D_v \leq 1$, this is negligible as it happens with probability less than 0.15%.

The results presented in figure 12 show that, when $D_v \leq 1$, BIV is robust to such noise. While a higher $D_v$ leads to lower performance, the impact is small compared to the effect of BIV. However, when $D_v = 2$, which is an arguably high level of noise and leads to bias as explained previously, the beneficial effect of BIV is significantly affected in BikeSharing, and completely disappears in UTKF.

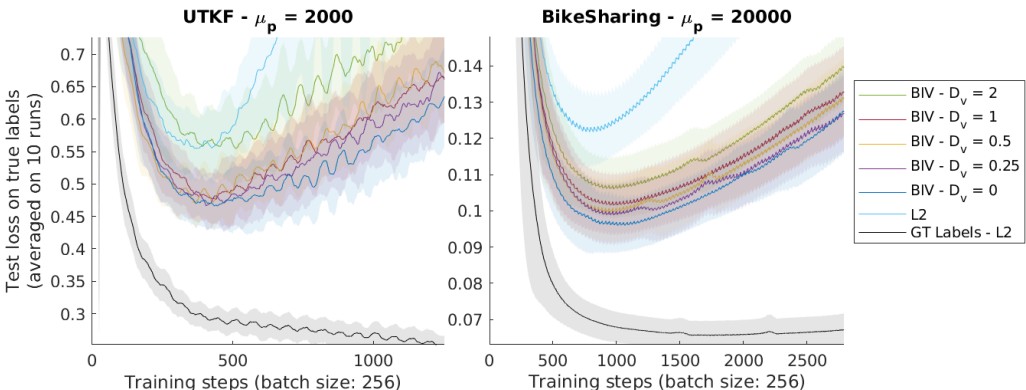

Figure 12: Robustness of BIV to noise in the variance with different disturbance coefficients $D_v$.

