# OpenReview forum: "Batch Inverse-Variance Weighting: Deep Heteroscedastic Regression"
_ICLR.cc/2021/Conference — Reject_

### Official Review · AnonReviewer2 · 2020-10-26
**Very well written paper, but I didn't find the contribution strong enough**

**Rating:** 4
**Confidence:** 4

**Review:**

##########################################################################

Paper Summary:
This work targets regression tasks with noisy labels, and proposes to incorporate knowledge about the variance of the gaussian noise corrupting the observed labels to weight the loss function, at training time. The proposed method is evaluated in a series of experiments involving deep networks trained according to the weighted loss function, and compared to a baseline method that omits training samples that have a label noise variance larger than a threshold.
Results indicate the proposed method is more robust to noisy labels when compared to alternatives that do not exploit the information on the noise affecting labels.

##########################################################################

Reasons for score:
On the one hand, the paper is extremely well written and somehow pedagogic, in that it provides compelling motivations for considering heteroscedasticity, and its possible sources, and general, intuitive descriptions of the proposed method before specialising them to the instance they evaluate. On the other hand, I think the prose lacks sufficient technical depth, on the model they use, its relation to “text book” material on heteroscedasticity, e.g. for Maximum Likelihood Estimation (MLE), and on the properties of the proposed method. The experimental evaluation, while representing a reasonable starting point, is not sufficient to fully understand the behaviour and the properties of the proposed method.
For these reasons, I think this work cannot be accepted as is.

##########################################################################

Positive points:

1) The editorial quality of this paper is high, and the overall motivations given to support the problem statement are compelling and well discussed. This also relates to the fundamental assumption underlying this work: access to privileged information, taking the form of knowledge of the stochastic noise variance affecting observed labels, at training time.

2) The proposed method appears to be well positioned w.r.t. the recent literature on statistical modelling with noisy labels, especially concerning neural network based methods. It is unfortunate though that the literature scan doesn’t cover well-known approaches to tackle heteroscedastic noise in simple linear models, or in general MLE frameworks, which may be considered text-book material.

3) The experimental evaluation considers two regression tasks on their respective UCI/Bike sharing, and UTK Face datasets, considering several variants of noise generation processes, affecting labels in different and sufficiently realistic manner.


##########################################################################

Negative points:
1) My main concern is the “thin” contribution of this paper. The technical details of the proposed method are not sufficiently developed. Drawing inspiration from Fisher information calls for an appropriate discussion on the likelihood model, its noise model, to begin with. Then, I think the relation of the proposed idea to simple linear models, for which heteroscedastic regression has been studied in great detail (e.g. [1], for a general reference), and for MLE (e.g., [2]), would become more clear and would give the opportunity for the authors to develop what are the merits of their proposed method. For example, the weighted least square method is very similar to what is proposed in this paper.
[1] Econometric analysis, Greene, William H, 2003, Pearson Education India
[2] Maximum Likelihood Estimators with Heteroscedastic Errors, G. R. Fisher, Review of the International Statistical Institute
Vol. 25,  1957
See also “Pattern Recognition and Machine Learning”, Bishop, 2006, (chapter 5 and 6)

2) The experimental evaluation is not sufficient to appreciate the virtues of the proposed method. For several noise distributions (including the additional ones considered in the supplement), the proposed method BIV does not seem to behave much better than the proposed baseline, that uses a simple threshold. Additionally, the figures are  cropped and do not allow to get a sense of what happens for all competing methods (Fig1 and Fig2). In Fig3, the figures report test loss and as there seems to be overfitting kicking in.
Also, it is mentioned the baseline method requires to set a cutoff parameter, but the proposed method also depends on a hyper parameter to optimise (done in the appendix). As a consequence, it is difficult to appreciate the main advantage of BIV w.r.t. the baseline.
Finally, in Fig.3, there are clear signs of overfitting: why did the authors suggest (end of Sec.3) that their work does not require regularisation?


#########################################################################

Main criticism:
I think, overall, the main criticism I have for this work is that the contribution is not sufficient. The main idea proposed in the paper fits sec 4.2, and it is based on well known results from text books. In eq.(5), the summation term is MLE with heteroscedasticity. The loss is scaled by a coefficient that collects statistics on the sample batch: if this batch is very small, or its elements not sufficiently diverse, I am afraid it could have a negative impact on the optimisation process (this is why, in the experiments, the authors chose a batch size of 256).
One possible way to overcome this criticism is to clarify the likelihood model, and compare the proposed method to existing approaches to address heteroscedastic Gaussian noise.
A possible advice would be to reduce (or move to the appendix) the discursive parts on heteroscedasticity, and the general formulations (e.g., sec. 2.2, sec. 4.1), and gain more space to explain how BIV is different from what is known.

Additional comments:
A note on experiments using the UTKFace dataset. In this case, the MLP used with 4 layers may be a bit “too simple”, in light of the high test loss on GT labels. Did you try with convolutional architectures, even simple ones?
Just to clarify, the test loss reported in the figures, as a function of training steps: what is the *test* batch size? Same as training batch size? The test loss is computer according to \sum_{k \in \text{test batch}} \mathcal{L}(f(\mathbf{x}_k, \theta), \tilde{y}_k) right?

---

> ### Author Response · Authors · 2020-11-17
> **Response to AnonReviewer2 - part 1/2**
>
> We thank the reviewer for their in-depth analysis of our work and their constructive remarks.
> As some elements came several times in the review, we took the liberty to summarize them in the two first paragraphs.
>
> **There is a need for clarification and discussion on the noise likelihood model.**
>
> We thank the reviewer for this remark. We indeed assume that the noise likelihood is Gaussian, and this asks for justification. We added some explanations in this respect at the beginning of section 2.1., before equation (1): Noise on measured or estimated values is often represented by Gaussian distribution, based on the central limit theorem as most noisy processes are the sum of several independent variables. Gaussian distributions are also practical, although it has some drawbacks as it can only represent uni-modal and symmetric noise models (Thrun et al., 2006).
>
> **There is a need for a thorough comparison to existing approaches to address heteroscedastic Gaussian noise (for simple linear models, weighted least squares with inverse variance weighting is the MLE solution)**
>
> Indeed, re-weighting using the inverse variance is a common strategy in regression, as it is the solution to MLE for Gaussian heteroscedastic noise. We hope to have answered this concern in the response to all reviewers posted above. We thank the reviewer for the references they provided, and have added some in the article.
>
> **For several noise distributions (including the additional ones considered in the supplement), the proposed method BIV does not seem to behave much better than the proposed baseline, that uses a simple threshold.**
>
> The performance of BIV with respect to the baseline (“cutoff”) is strongly dependent on the variance distribution $P(\sigma^2)$. When the “cutoff” can get a lot of low-noise data, such as the gamma distribution with alpha = .25 (fig 8 and 9), then the performances are very similar, and a lot better than L2. This is to be expected, as it resembles the binary uniform situation leading to Fig 2. When on the other side there is not a lot of support for low variances, BIV is doing similar to L2 (fig 10 with $\alpha > 1$) and so would “cutoff” (or worse, as its dataset would be very few data). However, in other cases when the distribution is uniform (fig. 7) or somehow balanced, as for gamma with $\alpha = 1$ (fig 3), BIV is clearly better than all cutoff and L2.
> What is important here is the consistency with which BIV is always at least better than the best of L2 and cutoffs, regardless of $P(\sigma^2)$. We have underlined this in section 6.3.
>
> **Also, it is mentioned the baseline method requires to set a cutoff parameter, but the proposed method also depends on a hyper parameter to optimise (done in the appendix). As a consequence, it is difficult to appreciate the main advantage of BIV w.r.t. the baseline.**
>
> The optimal value for $\epsilon$ depends on the distribution of labels in the dataset. If the latter is normalized (unit variance), then the optimal epsilon is between 0.01 and 0.1, regardless of $P(\sigma^2)$. As $\epsilon$ is a “minimal variance”, it is straightforward to scale to any other label distribution. This is very different for the cutoff parameter, which would need to be optimized for every dataset as it is dependent on $P(\sigma^2)$, as shown in our experiments. We agree that this advantage was not well explained, and we have clarified it in section 6.3.
>
> **Additionally, the figures are cropped and do not allow to get a sense of what happens for all competing methods (Fig1 and Fig2).**
>
> The competing methods that are cropped in Fig 1 and Fig 2 are the ones using the L2 loss, and they are actually overfitting because of the noise. We have taken advantage of having a bit more space to crop these figures a bit less, so that the viewer can have a better sense of what is happening.

---

> > ### Author Response · Authors · 2020-11-17
> > **Response to AnonReviewer2 - part 2/2**
> >
> > **Finally, in Fig.3, there are clear signs of overfitting: why did the authors suggest (end of Sec.3) that their work does not require regularisation?**
> >
> > Indeed, there is still overfitting with BIV, although it is less significant than with the competing methods. There is a confusion here. At the end of Sec. 3, we were comparing the loss function used in Kendall & Gal (2017) where log(sigma^2) is used to regulate the estimation of the data-related variance of the labels, which could be set to infinity otherwise as a trivial solution for their loss function. We have added a precision in the text to avoid confusing other readers.
> >
> > **The loss is scaled by a coefficient that collects statistics on the sample batch: if this batch is very small, or its elements not sufficiently diverse, I am afraid it could have a negative impact on the optimisation process (this is why, in the experiments, the authors chose a batch size of 256).**
> >
> > This is a very interesting remark. First, to clarify, we have not chosen a batch size of 256 for this reason. Instead, when determining the amount of noise for our experiments, we noticed that, as reported by Rolnick and al. (2017), dataset noise can be partly compensated for by larger batch sizes, as noisy labels roughly cancel out. In L2 loss indeed, the effect of noise was significantly lower when the batch size was higher. It is to ensure the best performances for our L2 baseline that we chose to keep 256. We have clarified this in section A.1.2.
> > With respect to the representativity of each batch: indeed, the differences between the normalization constants applied to each batch would be statistically higher for smaller batches. While this should impact the optimisation process by mitigating the effects of BIV, we do not expect that it makes it worse than the L2 or “cutoff” baselines with the same batch size. We plan however to provide the reviewer with experimental results in the following days, that will test this hypothesis and will be added in the Appendix with a mention in the text.
> >
> > **A note on experiments using the UTKFace dataset. In this case, the MLP used with 4 layers may be a bit “too simple”, in light of the high test loss on GT labels. Did you try with convolutional architectures, even simple ones?**
> >
> > There is a confusion here. For UTKFace, we did use the convolutional architecture of Resnet 18, as described in the appendix. We have tried several resnets and other simple convolutional architectures, and optimized the hyperparameters accordingly. Resnet18 was giving us the best results. We have made the precision in section 5. The GT labels have a test loss of 0.24, which represents a standard deviation of around 10 years old after de-normalizing, which is reasonable for this task.
> >
> > **Just to clarify, the test loss reported in the figures, as a function of training steps: what is the test batch size? Same as training batch size? The test loss is computer according to $\sum_{k \in \text{test batch}} \mathcal{L}(f(\mathbf{x}_k, \theta), \tilde{y}_k)$ right?**
> >
> > There is a confusion here. The test loss is computed with the ground-truth label values, in order to evaluate the performance of the model on its actual task (in most realistic situations, ground-truth test data would not be available, but this is not the point of the paper).
> > The loss is then $\sum_{k \in \text{test batch}} \mathcal{L}(f(\mathbf{x}_k, \theta), y_k)$, and the size of the test batch size is 256 too (even if it does not matter for the result, as it was computed on the whole test set after each training step). We have clarified this in the new section 5.2.

---

> > > ### Author Response · Authors · 2020-11-24
> > > **Results about the impact of batch size**
> > >
> > > We would like to notify the reviewer that we have updated our manuscript with new results, as announced in our previous response.
> > >
> > > We ran experiments with batch sizes of 16 to 256 and we show that the size of the mini-batch does not have a significant impact on the optimization process compared to L2 learning with the same batch size. These results are mentioned in section 6.4 and more details can be found in the appendix B.4.1. We hope that they address the concern raised by the reviewer.

---

### Official Review · AnonReviewer1 · 2020-10-27
**This paper proposes a reweighed loss function for robust regression model training against label noise. The weight value of each training instance is determined by prior knowledge about the noise generation process. Results confirm empirically the merit of the algorithmic design**

**Rating:** 4
**Confidence:** 5

**Review:**

In this paper, a reweighting technique is proposed to suppress the impact of heteroscedastic label noise in regression model training. The objective function of the regression model training process is composed of a weighted combination of instance-wise training loss. The instance-wise weight is determined by the estimated noise variance based on prior information of the label generation process. The weighting formulation is inspired by the best possible estimator of noisy measurements reaching the Cramer-Rao bound.

The paper is clearly written. It explains well the problem definition and the methodological formulation. However, we think the innovation in this work is limited. The downside of this paper is as follows:
1. It is a strong and usually impractical assumption to know a priori knowledge of label noise in the regression model training process. Especially, in the proposed method, the estimate of the noise variance needs to be accurate enough, so as to help suppress the noise impact accordingly. It is better to incorporate jointly the learning of noise distribution and the regression/classification
model, so as to optimize the tolerance against the data-dependent noise.
2. It only considers the noisy learning process for regression. However, in classification scenarios, noise (with respect to labels) is usually presented in the form of label flipping. The proposed reweighing technique is not directly applicable in that case. Please refer to the following work for further reading:
Learning with Noisy Labels, Nagarajan Natarajan, Inderjit S. Dhillon, Pradeep Ravikumar, and Ambuj Tewari, NIPS 2013.

---

> ### Author Response · Authors · 2020-11-17
> **Response to AnonReviewer1**
>
> We thank the reviewer for their constructive review.
>
> **It is a strong and usually impractical assumption to know a priori knowledge of label noise in the regression model training process.**
>
> We hope to have answered this concern in the response to all reviewers posted above.
>
> **Especially, in the proposed method, the estimate of the noise variance needs to be accurate enough, so as to help suppress the noise impact accordingly.**
>
> Indeed, further studies on the robustness of BIV to noise in the noise variance estimation should be made to ensure that this method is applicable in real world cases. We are planning to add new experimental results in this respect in the following days. We thank the reviewer for this suggestion!
>
> **It is better to incorporate jointly the learning of noise distribution and the regression/classification model, so as to optimize the tolerance against the data-dependent noise.**
>
> This strategy is indeed used in many frameworks, such as Kendall et al., 2017. The main problem, as correctly underlined by the reviewer, is that it assumes that the noise is data-dependent, which disregards the noise due to the label generator. A mixture of both methods could be interesting however when both kinds of noise are present. We thank the reviewer for their comment, as it will fuel our future reflection on how to improve this work.
>
> **Please refer to the following work for further reading: Learning with Noisy Labels, Nagarajan Natarajan, Inderjit S. Dhillon, Pradeep Ravikumar, and Ambuj Tewari, NIPS 2013.**
>
> We have added the reference, thank you very much! It is actually very interesting, as this work assumes the knowledge of the probability \rho of mislabeling, which is very close to our assumption of knowing the variance. As it is applied in binary classification with the same \rho for each category, the authors are able to optimize these two hyperparameters using cross-validation, which is of course not possible in our case where each sample has its own noise distribution.
>
> **It only considers the noisy learning process for regression. However, in classification scenarios, noise (with respect to labels) is usually presented in the form of label flipping. The proposed reweighing technique is not directly applicable in that case.**
>
> There is a lot of work in the case of classification, including the reference the reviewer provided, which takes into account the probability of label flipping, and which are not applicable to regression. In our case, we try to solve the problem in regression, where a lot less work has been made in the field of learning with noisy labels.

---

> > ### Author Response · Authors · 2020-11-24
> > **Results about importance of variance accuracy**
> >
> > We would like to notify the reviewer that we have updated our manuscript with new results, as announced in our previous response.
> > We show that BIV is robust to moderate to high levels of noise in the variance. These results are mentioned in part 6.4 and the details are in the appendix B.4.2.
> > We hope this addresses the concern raised by the reviewer with respect to the assumption that the estimate of the variance is accurate enough.

---

### Official Review · AnonReviewer4 · 2020-10-29
**Important References are Missing; Novelty is Unclear.**

**Rating:** 3
**Confidence:** 3

**Review:**

The paper propose to address the heteroscedastic regression problem using deep neural networks. It assumes the variance of heteroscedastic noise is known as privileged information and suggests to reweight the samples by their noise variance in the loss.

The major issue to me is the lack of novelty. Heteroscedastic regression is a classic problem in statistics. And reweighting using the inverse variance is a textbook method. See Chapter 10 of

http://www.stat.cmu.edu/~cshalizi/ADAfaEPoV/ADAfaEPoV.pdf

This paper failed to cite any relevant reference and clarify the novelty.

To apply the method to a deep learning setting, some interesting problem can be how to estimate the variance with deep network in a reliable way (this was done previously using classic models). However, this paper did not tackle this harder (and more interesting) problem. Instead, it assume the variance is simply given during training. This is not very realistic in real world setting. The experiments are all synthetic and are not particularly convincing.

Finally, the paper claims a lot of connection with privileged information (LUPI). But I found it hard to consider this variance a similar concept as privileged information, which is realistic and interesting.

---

> ### Author Response · Authors · 2020-11-17
> **Response to AnonReviewer4**
>
> We thank the reviewer for their constructive review.
>
> **Heteroscedastic regression is a classic problem in statistics. And reweighting using the inverse variance is a textbook method. See Chapter 10 of http://www.stat.cmu.edu/~cshalizi/ADAfaEPoV/ADAfaEPoV.pdf**
>
> We hope to have answered this concern in the response to all reviewers posted above. We thank the reviewer for the reference, which we have added in the article.
>
>
> **Instead, it assumes the variance is simply given during training. This is not very realistic in real world setting.**
>
> We also hope to have answered this concern in the response to all reviewers posted above.
>
>
> **The experiments are all synthetic and are not particularly convincing.**
>
> Given the fact that no regression dataset currently makes the uncertainty available, we were not able to make non-synthetic experiments. However, we do think that our results show convincingly that the method does perform significantly better than L2 and consistently better than a baseline. We are conscious that more detailed experiments, such as on the robustness to errors in the variance estimation, can be made, and we plan to add them to the paper in the following days.
>
>
> **Finally, the paper claims a lot of connection with privileged information (LUPI). But I found it hard to consider this variance a similar concept as privileged information, which is realistic and interesting.**
>
> On further review based on this comment, we agree that the concepts of LUPI and variance heteroscedastic regressions are different, although interesting parallels can be made at the problem definition and objective levels. We have removed the direct references to LUPI both in the title and in several places in the text, and confined the comparison to section 2.2. We thank the reviewer for underlining that the connection claim was too strong.
>
>
> **To apply the method to a deep learning setting, some interesting problem can be how to estimate the variance with deep network in a reliable way (this was done previously using classic models). However, this paper did not tackle this harder (and more interesting) problem.**
>
> Indeed, the problem of reliably estimating the variance with deep networks is extremely interesting and has seen a lot of recent attention (Kendall & Gal, 2017; Gal & Ghahramani, 2016; Peretroukhin, 2019). The method we propose has a completely different objective however, which is to improve learning when confronted with noisy labels. We believe that the combination of both problems will be beneficial in complex neural architectures where the output of neural networks is a noisy label used for training another network.

---

> > ### Author Response · Authors · 2020-11-24
> > **Results about the robustness of BIV**
> >
> > We would like to notify the reviewer that we have updated our manuscript with new results, as announced in our previous response.
> >
> > We show that BIV is robust to moderate to high levels of noise in the variance. We also show that the number of samples in a mini-batch is not critical to the performance of BIV. These results are mentioned in part 6.4 and the details are in appendix B.4.
> >
> > We hope that these results help making the experiments more convincing, as this was a concern raised by the reviewer.

---

### Author Response · Authors · 2020-11-17
**Response to all reviewers**

We thank all the reviewers for their very valuable feedback. It is precious in the process of improving this article as well as the research project it stems from.

All the reviewers were concerned with the novelty of the method. Indeed, we had presented our algorithm through an Fisher information perspective, while the reviewers rightly pointed out that our method has a lot in common with the classic inverse-variance strategies for linear heteroscedastic regression, which achieves maximum likelihood estimation of the model parameters when the noise is Gaussian. We updated the article accordingly, replacing the paragraph about Fisher information by one on linear heteroscedastic regression and citing the references presented by the reviewers, in section 2.3.
There, we also underlined the aspects of this work which are novel compared to the text-book methods: it is the adaptation of such a strategy for the stochastic gradient descent methods applied to neural networks. Applying inverse-variance reweighting in the loss function directly is prone to problems: (1) when in presence of near-ground truth samples (2) when the learning rate needs to be controlled (3) when pre-processing the data cannot be done such as in continual learning. To the best of our knowledge, this is the first work proposing a solution to such issues, with both the role of epsilon and the batch-based normalization.

Another common concern is the realism of the assumption of knowing the label variance during training. While it is definitely not an assumption that holds for every dataset, we argue that having labels that are not estimated is, on the contrary, not an assumption that holds for all use cases. From robotics to studies on population to simulation, there are a lot of cases in which labels come with uncertainty, and for which this uncertainty is quantified. The question we address in this paper is: when such information is available, how can we use it in an optimal way? While this is not common in current datasets for regression, we hope that, if new methods such as the one we propose prove that such information is actually valuable, more datasets will take the extra step of including it. To answer these concerns, we have thus strengthened part 2.1 with more cases and the example of a dataset already including a confidence score.

Each reviewer had particular questions and comments that we answered individually. Overall, we made several changes in the paper, which can be found in blue in the new version.

Please note that, for a better statistical significance of our results, we have repeated the experiments and now plot the average over 10 runs instead of 3. The overall results have not changed.

We are also planning to release additional experimental results in the following days. Based on the remarks from two reviewers, we will study the robustness of our method to noise in the variance estimation, as well as the impact of smaller batch sizes.

We would appreciate it if the reviewers can confirm that their concerns have been addressed and, if so, reconsider their assessment. Once again, we thank them for their precious time and comments.

---

### Decision · Program_Chairs · 2021-01-07
**Final Decision**

**Decision:**

Reject

**Comment:**

The manuscript presents a deep network approach for heteroscedastic regression problem. It assumes the variance of heteroscedastic noise is known as privileged information and suggests to reweight the samples by their noise variance in the loss.

Three reviewers agreed that the manuscript is not ready for publication. The major issue is the lack of novelty. Heteroscedastic regression is a classic problem in statistics. And reweighting using the inverse variance is a textbook method.

R2 and R4 confirmed that they have read author response. The rebuttals are useful to clarify some points, especially related to experimental settings and results. However, they are not convinced by the authors' argument on novelty and whether the assumption is realistic.